# A suite of PCR-LwCas13a assays for detection and genotyping of *Treponema pallidum* in clinical samples

Wentao Chen [1,2,10], Hao Luo [1,2,10], Lihong Zeng[1,2], Yuying Pan[1,2], Jonathan B. Parr [3,11], Yinbo Jiang[1,2], Clark H. Cunningham[3], Kelly L. Hawley[4,5,6,11], Justin D. Radolf[5,6,7,8,9,11], Wujian Ke[1,2], Jiangli Ou[1,2], Jianjiang Yang[1,2], Bin Yang[1,2,11] ✉ & Heping Zheng [1,2,11] ✉

The performance of commonly used assays for diagnosis of syphilis varies considerably depending on stage of infection and sample type. In response to the need for improved syphilis diagnostics, we develop assays that pair PCR pre-amplification of the *tpp47* gene of *Treponema pallidum* subsp. *pallidum* with CRISPR-LwCas13a. The PCR-LwCas13a assay achieves an order of magnitude better analytical sensitivity than real-time PCR with equivalent specificity. When applied to a panel of 216 biological specimens, including 135 clinically confirmed primary and secondary syphilis samples, the PCR-LwCas13a assay demonstrates 93.3% clinical sensitivity and 100% specificity, outperforming *tpp47* real-time PCR and rabbit-infectivity testing. We further adapt this approach to distinguish *Treponema pallidum subsp. pallidum* lineages and identify genetic markers of macrolide resistance. Our study demonstrates the potential of CRISPR-based approaches to improve diagnosis and epidemiological surveillance of syphilis.

*Treponema pallidum* subspecies *pallidum* (*TPA*) is the causative agent of syphilis, a resurgent sexually transmitted disease worldwide[1,2]. Syphilis can progress through multiple stages and present with a wide variety of clinical manifestations[2]. Laboratory testing is essential for syphilis diagnosis, but existing assays have limited clinical sensitivity, specificity, or both[2–4]. There is no single diagnostic test with sufficient sensitivity and specificity to identify all stages of disease[3,4]. Because neither *TPA* cultivation[5] nor rabbit-infectivity testing[6,7] to isolate *TPA* strains from patients can be done in routine clinical practice, clinical diagnosis currently relies upon serological testing[2]. Assays employing nontreponemal and treponemal serological tests show high sensitivity

(95%) for secondary and later stages of syphilis[8] but reduced sensitivity for primary syphilis[4,9,10]. Lifelong positivity of treponemal and non-treponemal tests (the 'serofast' state) further complicates the interpretation of serological results and response to syphilis treatment[11–15].

Established methods for direct detection of *TPA*, such as darkfield microscopy (DFM)[16], direct fluorescence antibody (DFA) testing[17] and nucleic acid amplification test (NAAT)[18,19], are often used to complement serological testing but have distinct limitations. DFM and DFA are heavily reliant upon the technologist's expertize, and the performance of these methods and NAAT vary based on specimen type[17–21]. Though NAAT is well-established and accurate, the sensitivity of various NAAT

[1]Dermatology Hospital, Southern Medical University, Guangzhou, P. R. China. [2]Guangzhou Key Laboratory for Sexually Transmitted Diseases Control, Guangzhou, P. R. China. [3]Division of Infectious Diseases, Department of Medicine, and Institute for Global Health and Infectious Diseases, University of North Carolina, Chapel Hill, NC, USA. [4]Division of Infectious Diseases, Connecticut Children's, Hartford, CT, USA. [5]Department of Medicine, UConn Health, Farmington, CT, USA. [6]Department of Pediatrics, UConn Health, Farmington, CT, USA. [7]Department of Molecular Biology and Biophysics, UConn Health, Farmington, CT, USA. [8]Department of Genetics and Genome Sciences, UConn Health, Farmington, CT, USA. [9]Department of Immunology, UConn Health, Farmington, CT, USA. [10]These authors contributed equally: Wentao Chen, Hao Luo. [11]These authors jointly supervised this work: Jonathan B. Parr, Kelly L. Hawley, Justin D. Radolf, Bin Yang, Heping Zheng. ✉e-mail: yangbin1@smu.edu.cn; zhengheping@smu.edu.cn

assays ranges widely in different stages of disease and specimen types and is problematic for use in neurosyphilis[22]. For mucocutaneous lesion exudates, reported sensitivities of NAATs range from 75–95% in primary syphilis and 20–86% in secondary syphilis[20]. NAATs applied to whole blood reach sensitivities of only 13% in primary syphilis and 38–64% in secondary syphilis, respectively[18,23]. Given recent dramatic increases in the incidence of syphilis globally, the development of highly sensitive and specific *TPA* assays should be considered a priority.

Improved assays also are needed to clarify syphilis molecular epidemiology and disease pathogenesis[24]. The original and widely used treponemal typing system (CDC) could not distinguish between the dominant Nichols- and SS14-like clades until the addition of the *tp0548* locus to the enhanced typing system (ECDC)[25,26]. However, newer multilocus sequence typing (MLST) and whole-genome sequencing approaches are providing insights into the genetic diversity of *TPA* strains circulating within hyper-endemic populations[27–34]. Azithromycin is an alternative antimicrobial for treatment of syphilis in persons with significant allergic reactions to penicillin G. However, the prevalence of *TPA* harboring mutations associated with resistance to macrolide antibiotics (A2058G and A2059G *23 S rRNA* mutations) has increased rapidly, and these mutations are now fixed in many localities[35–39]. In the absence of robust in vitro cultivation systems for *TPA* strains in clinical samples, assessment of resistance relies upon molecular genotyping.

New technologies such as clustered, regularly interspaced palindromic repeat (CRISPR)-based diagnostic assays provide exciting avenues to overcome current limitations of syphilis diagnostic tests[40]. Methodologies such as SHERLOCK, HOLMES, and DETECTR that leverage the highly sensitive and specific activity of CRISPR-Cas family proteins have now been applied to diagnose numerous organisms, *e.g.* SARS-CoV-2[41,42], *Zika* virus[40,43], *Dengue* virus[43,44], *Plasmodium* species[45,46] and *Pseudomonas aeruginosa*[47], and to distinguish single nucleotide variants (SNVs) for genotyping and drug-resistance monitoring[40,43–45,48,49]. In the present study, we developed and validated an assay for *TPA* that combines polymerase chain reaction (PCR) pre-amplification, CRISPR-RNA guided pairing and LwCas13a cleavage activity. This assay exhibits robust performance across different types of biological samples and improves sensitivity by an order of magnitude compared to PCR alone. We further adapted this approach for *TPA* lineage identification and macrolide resistance genotyping. Together, this suite of molecular assays demonstrates the potential of CRISPR-based approaches for improved diagnosis and epidemiologic surveillance for syphilis.

## Results

### Highly sensitive and specific detection of *Treponema pallidum* DNA by PCR-LwCas13a

We developed a *TPA* DNA diagnostic assay by pairing PCR and CRISPR-LwCas13a detection; the latter is based on cleavage of an RNA reporter following RNA-guided (crRNA) target recognition. In brief, nucleic acids are extracted from a clinical sample, and the target gene is amplified by PCR including one primer tagged with a T7 promotor sequence. PCR products then are detected in a reaction mixture containing T7 RNA polymerase, LwCas13a, a target-specific crRNA, and an RNA reporter that fluoresces when cleaved (Fig. 1). The *TPA* 47 kDa lipoprotein (*tpp47/tp0574*) and DNA polymerase I (*polA/tp1021*) genes, the two most common targets for PCR detection of *TPA*, exhibit negligible differences in detection efficiency[20,50]. We selected *tpp47* as the target of the PCR-LwCas13a assay because, in our hands, it produced a higher level of fluorescence signal compared to *polA* (Fig. S1).

To evaluate the analytical sensitivity of the assay, we synthesized and purified double-stranded DNA containing *tpp47* for a dilution series in which mock clinical samples contained 1 ng human DNA (to simulate human DNA background in clinical samples). The concentrations of diluted aliquots were confirmed by digital droplet PCR (ddPCR) (Fig. S2). The PCR-LwCas13a assay detected all replicates with >$10^0$ copies of *tpp47* per reaction, equivalent to an analytical sensitivity of approximately 1 spirochete/reaction, one order of magnitude more sensitive than TaqMan PCR for *tpp47* (Fig. 2a). We compared our assay to SHERLOCK, another sensitive CRISPR-based diagnostic that combines isothermal recombinase polymerase amplification (RPA) and CRISPR-LwCas13a detection[40,42,43,45]. We found the performance of SHERLOCK for *TPA tpp47* detection to be inferior to that of PCR-LwCas13a assay. The PCR-LwCas13a assay achieved an order of magnitude better analytical sensitivity than SHERLOCK (single-copy versus 10 copies/reaction, respectively) (Fig. 2a and S3a) determined by comparison of both methods using serially diluted clinical samples and a small sample set (Fig. S3b and Supplementary Table 1). The PCR-LwCas13a assay also exhibited higher sensitivity than *tpp47*-based nested PCR for 30 whole blood samples (83% vs 63%; see Supplementary Table 2).

The analytical specificity of the PCR-LwCas13a assay was validated by testing a panel of 10 genital microorganisms (*Herpes simplex virus*−1,

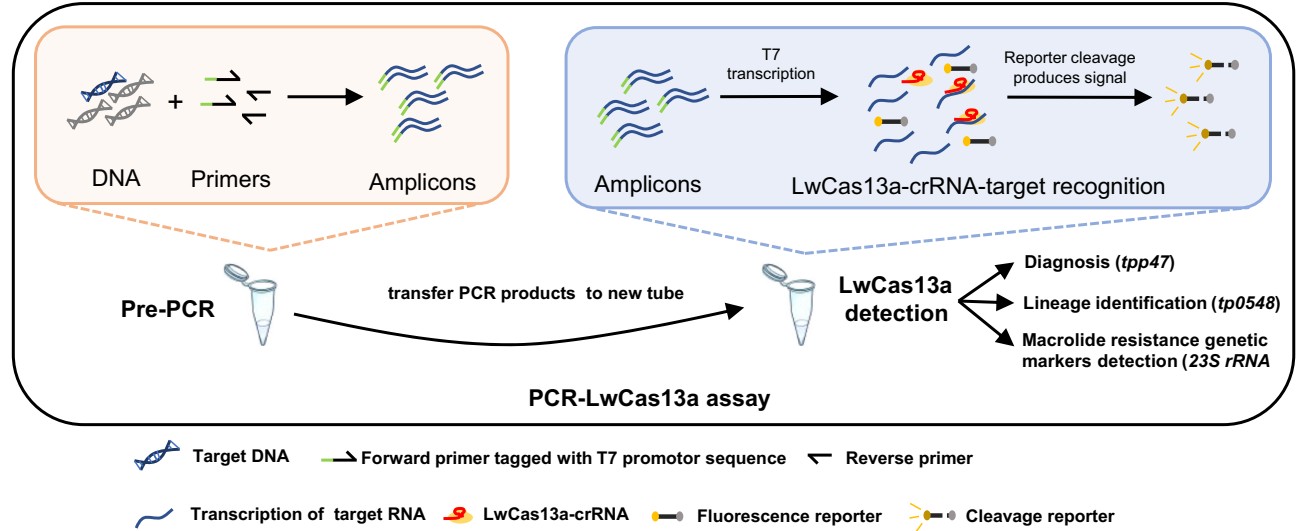

**Fig. 1 | Schematic of the PCR-LwCas13a assay for detection and genotyping of *Treponema pallidum*.** The target is pre-amplified by PCR with DNA as the input. PCR products are transferred to and detected in a reaction mixture containing T7 RNA polymerase, LwCas13a, target-specific crRNA, and an RNA reporter that fluoresces at Ex/Em = 490 nm/520 nm when cleaved. Three targets of *TPA* were detected in separate reactions in triplicate for diagnosis, identification of lineage and macrolide resistance genetic markers, respectively.

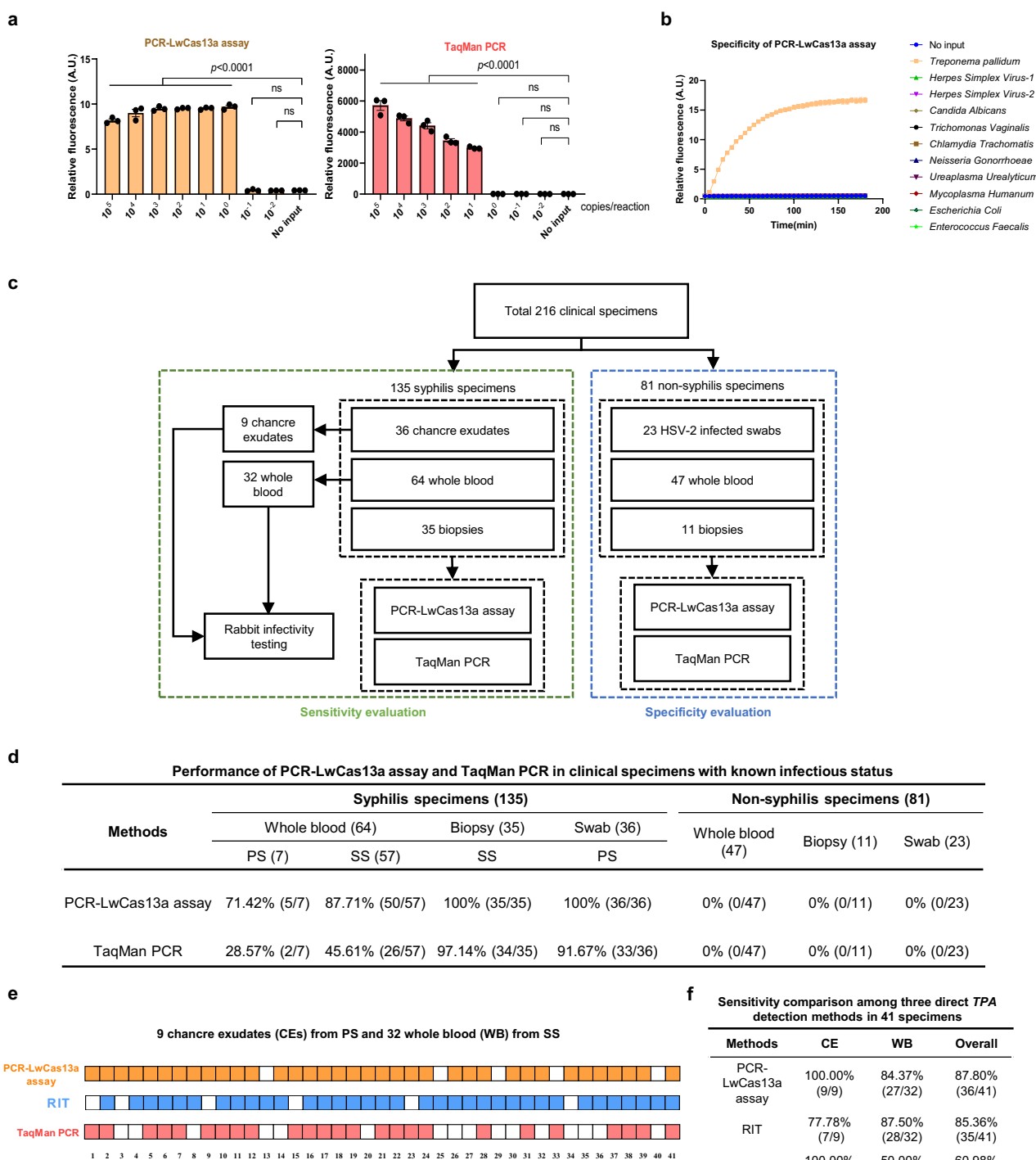

**Fig. 2 | Robust performance of the PCR-LwCas13a assay for detection of _Treponema pallidum_ in clinical specimens. a** The PCR-LwCas13a assay exhibits excellent detection of the _tpp47_ gene and is an order of magnitude more sensitive than TaqMan PCR. **b** Evaluation of the specificity of the PCR-LwCas13a assay by testing DNA from a group of 10 genital microorganisms in 180 min of kinetics analysis. **c** Samples from 135 syphilis and 81 non-syphilis patients (216 total) were used to determine the clinical sensitivity and specificity of the PCR-LwCas13a assay in a parallel comparison with TaqMan PCR. Rabbit-infectivity testing (RIT) was performed on 9 chancre exudates and 32 whole blood samples from secondary syphilis. **d** Performance of the PCR-LwCas13a assay and TaqMan PCR with clinical specimens. **e** Comparison of direct detection with whole blood from 32 patients with secondary syphilis and chancre exudates from nine patients with primary syphilis. Squares with color (orange for PCR-LwCas13a assay, blue for rabbit-infectivity test, and red for TaqMan PCR) represent positive signals, while white squares represent negative results. Darkfield microscopy of testicular extracts was used to determine RIT results. **f** Comparison of the sensitivities of direct detection methods. Abbreviations: A.U. Arbitrary units, PS Primary syphilis, SS Secondary syphilis, CE Chancre exudate, WB Whole blood, RIT rabbit-infectivity test. _n_ = 3 technical replicates; two-tailed Student _t_-test was used to analyze the statistical significance; _p_-value was labeled in the figure, ns = not significant; error bars represent mean ± SEM. Source data are provided as a Source Data file.

*Herpes simplex virus−2, Candida albicans, Trichomonas vaginalis, Chlamydia trachomatis, Neisseria gonorrhoeae, Ureaplasma urealyticum, Mycoplasma humanum, Escherichia coli, Enterococcus faecalis*) at high DNA concentrations (Fig. 2b and Supplementary Table 3). The PCR-LwCas13a assay can be optimized for rapid performance. As shown in Fig. S4, the assay could detect all serial dilutions of *tpp47* dsDNA within a 75 min reaction time (60 min pre-PCR pairing with 15 min LwCas13a). We observed a correlation ($R^2 = 0.897$) of copy numbers of *TPA tpp47* synthetic dsDNA with detected fluorescence under 15 min LwCas13a detection (Fig. S5), suggesting the potential of the PCR-LwCas13a assay for DNA quantitation.

### Robust performance of the PCR-LwCas13a assay for detection of *Treponema pallidum* in clinical specimens

To determine the clinical sensitivity and specificity of the PCR-LwCas13a assay, we collected 216 clinical specimens as shown in Fig. 2c. For evaluation of clinical sensitivity, we examined 135 syphilis specimens with known infection status based on clinical diagnosis and serological results (Supplementary Table 4). For comparison, the *tpp47* PCR-LwCas13a assay and TaqMan PCR were conducted in parallel. The overall sensitivities for *TPA* detection were 93.33% (95% CI: 87.72–96.91%) for the PCR-LwCas13a assay and 70.37% (95% CI: 61.91–77.92%) for TaqMan PCR. Among different specimen types, the PCR-LwCas13a assay exhibited greater sensitivity than TaqMan PCR in whole blood from patients with primary (PS, 71.42% vs. 28.57%) and secondary syphilis (SS, 87.71% vs. 45.61%), skin biopsies from SS (100% vs. 97.14%), and genital swabs from PS (100% vs. 91.67%). Notably, the sensitivity of PCR-LwCas13a assay for whole blood was substantially greater than TaqMan PCR (Fig. 2d).

To determine the clinical specificity of the PCR-LwCas13a assay, we examined 81 non-syphilis samples, including 47 whole blood, 11 skin biopsies and 23 genital swabs. Since the differential diagnosis of genital chancres often includes HSV-2, the 23 genital swabs were all from HSV-2 infected patients based on clinical diagnosis and laboratory test results. Specificities of 100% (95% CI: 94.9–100.0%) were observed for both PCR-LwCas13a assay and TaqMan PCR against all three specimen types (Fig. 2d).

Rabbit-infectivity testing (RIT) has long been considered the gold standard for direct detection of *TPA*, but it cannot be used routinely in clinical settings due to its high cost and requirement for specialized facilities and experienced personnel[6,7]. To further evaluate the performance of the PCR-LwCas13a assay, we compared it to Taqman PCR and RIT for 41 of the above-described specimens (32 whole blood from secondary syphilis and 9 chancre exudates) (Fig. 2e and Supplementary Table 4). The PCR-LwCas13a assay, RIT and TaqMan PCR exhibited total sensitivities of 87.80%, 85.36% and 60.98%, respectively. The PCR-LwCas13a and TaqMan PCR assays exhibited sensitivities of 100% for chancre exudates compared to 77.78% for RIT. For whole blood, PCR-LwCas13a assay exhibited similar sensitivity to RIT (84.37% vs. 87.50%), while TaqMan PCR was far less sensitive (50.00%) (Fig. 2f). The comparable sensitivity of PCR-LwCas13a to RIT, coupled with its superiority over conventional TaqMan PCR, demonstrates its potential utility in a clinical setting.

### Adapting PCR-LwCas13a for identification of *Treponema pallidum* lineage

*TPA* strains circulating worldwide have been divided into SS14 and Nichols lineages based on multi-locus analysis and genomic sequences[31,32,34,51]. Currently, *TPA* genotyping relies on sequence-based approaches[27,28,52]. We next adapted our PCR-LwCas13a assay for genotyping of *TPA*. Briefly, to enable the PCR-LwCas13a assay to distinguish the two major *TPA* clades, we designed two distinct crRNAs targeting a variant region in *tpO548*, a locus widely used for ECDC and MLST genotyping of *TPA* strains (Fig. 3a and S6a)[26,27,51]. As shown in

Fig. 3b–d, each genotyping crRNA successfully identified the corresponding *TPA* clade without any cross-reactivity. In dilution series using synthesized *tpO548* DNA for the Nichols and SS14 strains, the assay yielded genotyping data for samples containing as few as 10 copies per reaction (Fig. 3e). To validate this genotyping assay, we compared the traditional molecular typing approach (PCR followed by Sanger sequencing) with the PCR-LwCas13a assay using DNA extracted from ten skin biopsy samples and their corresponding rabbit-passaged isolates (twenty total samples). Results of the PCR-LwCas13a genotyping assay matched the Sanger sequencing clade assignments for all SS14 and Nichols clade samples tested (Fig. 3f), thereby confirming that the PCR-LwCas13a assay is capable of *TPA* genotyping. When applied to a larger set of 33 *tpp47*-positive clinical samples, the PCR-LwCas13a assay successfully identified *TPA* lineages for 32, a sensitivity of 96.97% (Fig. 3g).

### Adapting the PCR-LwCas13a assay for macrolide resistance genotyping of *T. pallidum* strains

To adapt the PCR-LwCas13a assay for identification of *23S rRNA* mutations associated with azithromycin resistance[35–39], we first screened crRNAs targeting the *23S rRNA* gene to differentiate wildtype and mutant strains based on their collateral cleavage activities and specificity ratios (Fig. 4a and S7a); we selected the 26 nt-long crRNA#2 for further testing (Fig. S7a–c).

The analytical sensitivity of PCR-LwCas13a assay for *23S rRNA* mutation identification was evaluated by a dilution series of *23S rRNA* dsDNA (A2058G mutation). The assay detected all replicates with greater than an estimated $7.22 \times 10^1$ copies/reaction of *23S rRNA* dsDNA (A2058G mutation) (Fig. 4b). The analytical specificity of PCR-LwCas13a assay was validated using DNAs from azithromycin-sensitive *TPA*, *TPA* with known azithromycin resistance, and a panel of genital microorganisms (Supplementary Table 3). We observed collateral cleavage activity only for the *23S rRNA* mutations (A2058G and A2059G) (Fig. 4c). To further validate these results, we tested ten clinical specimens by Sanger sequencing and the PCR-LwCas13a assay in parallel; synthesized wildtype *23S rRNA* dsDNA was included as a negative control. The PCR-LwCas13a assay accurately identified the *23S rRNA* mutations from clinical samples, matching the results obtained by Sanger sequencing (Fig. 4d).

## Discussion

Improved diagnostics are critical for efforts to address the alarming global increase in syphilis incidence, including adverse outcomes of pregnancy, seen during the past 20 years[1,2]. Herein, we describe a suite of direct detection assays for *TPA* that combine PCR pre-amplification and LwCas13a-based detection to improve sensitivity by an order of magnitude compared to real-time PCR alone using *tpp47* as the target gene. We validated these assays using a large bank of 216 clinical samples recently collected from subjects with known infection status and from diverse anatomical sites. We further demonstrated their utility for *TPA* detection, lineage identification, and detection of azithromycin-resistance.

The reduced sensitivity of serological assays for primary syphilis, along with the complicated interpretation of serological testing for diagnosis and response to treatment, underscore the need for improved methods for direct detection of *TPA*. A recent report demonstrating high loads of *TPA* DNA in the saliva of syphilis patients that is reduced following treatment further highlights the potential utility of direct detection[53,54]. The limited sensitivity and specificity of currently available methods for direct detection of *TPA* precludes use of a single method for all specimen types. DFM lacks specificity in oral lesions or saliva, because it fails to distinguish *TPA* from *T. denticola* and other oral treponemes[3]. Real-time PCR also loses sensitivity when applied to whole blood[18,20,23]. In the present study, the sensitivity of our PCR-LwCas13a assay targeting *TPA tpp47* across our testing sample set

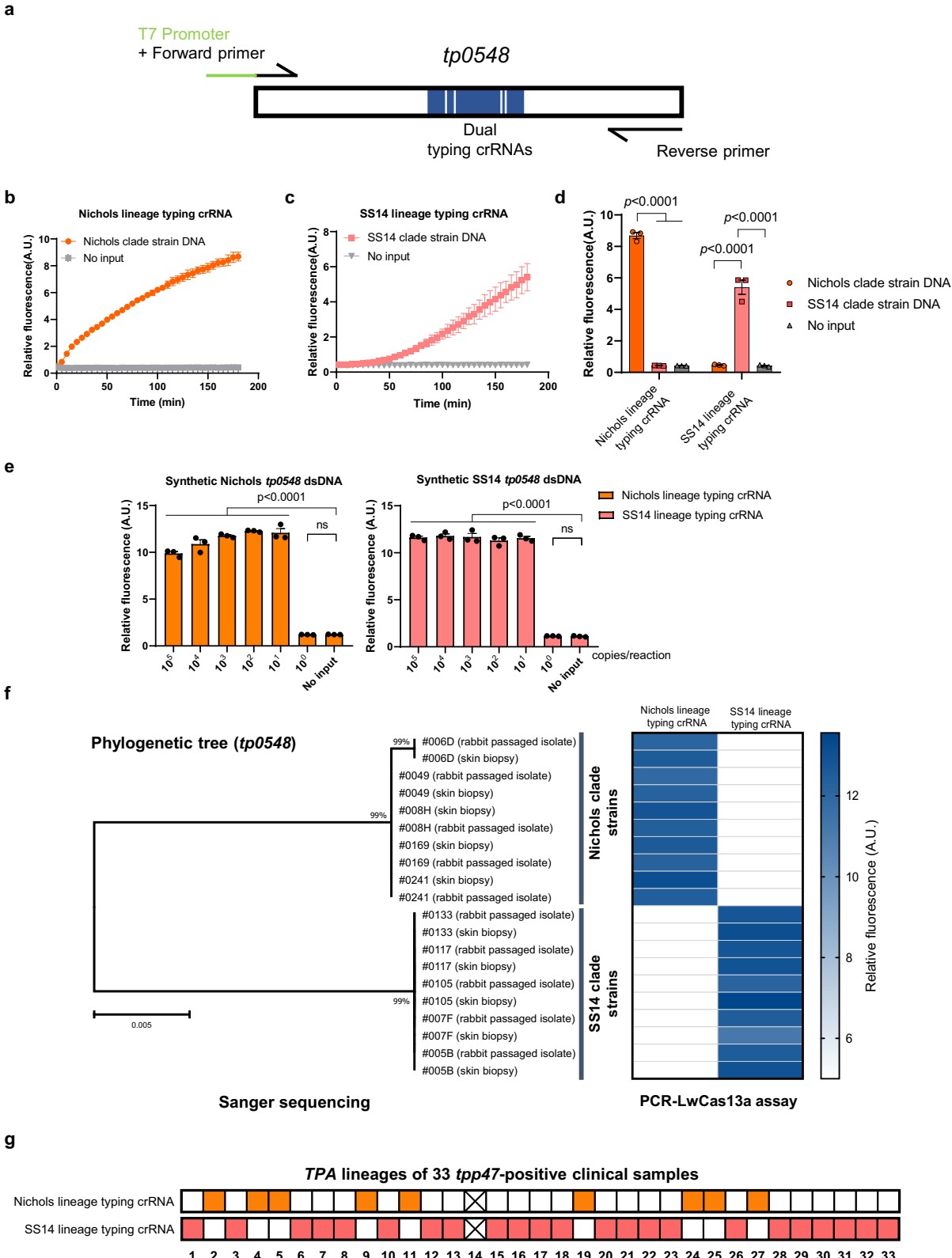

**Fig. 3 | Adapting the PCR-LwCas13a assay for determination of _Treponema pallidum_ lineage. a** Schematic of PCR-LwCas13a assay for determination of _TPA_ lineage. **b**–**d** Fluorescence measurement of Nichols and SS14 clade strains by PCR-LwCas13a-based genotyping. **e** Dilution experiment to assess the limits of detection of the _tp0548_-based PCR-LwCas13a genotyping assay. **f** Ten skin biopsy samples and their corresponding rabbit-passaged _TPA_ isolates were genotyped using the PCR-LwCas13a assay and Sanger sequencing. The phylogenetic tree for _tp0548_ sequences was constructed using MEGA-X. The tree scale bar indicates the average number of nucleotide substitutions per site. **g** PCR-LwCas13a assay identified the _TPA_ lineages for 32 of 33 _tpp47_-positive clinical samples. Squares with color represent positive signals, while white squares represent negative results. Squares with cross represents no positive signals detected by either Nichols or SS14 lineage crRNAs. Abbreviations: A.U. Arbitrary units. _n_ = 3 technical replicates; two-tailed Student _t_-test was used to analyze the statistical significance; _p_-value was labeled in the figure, ns = not significant; error bars represent mean ± SEM. Source data are provided as a Source Data file.

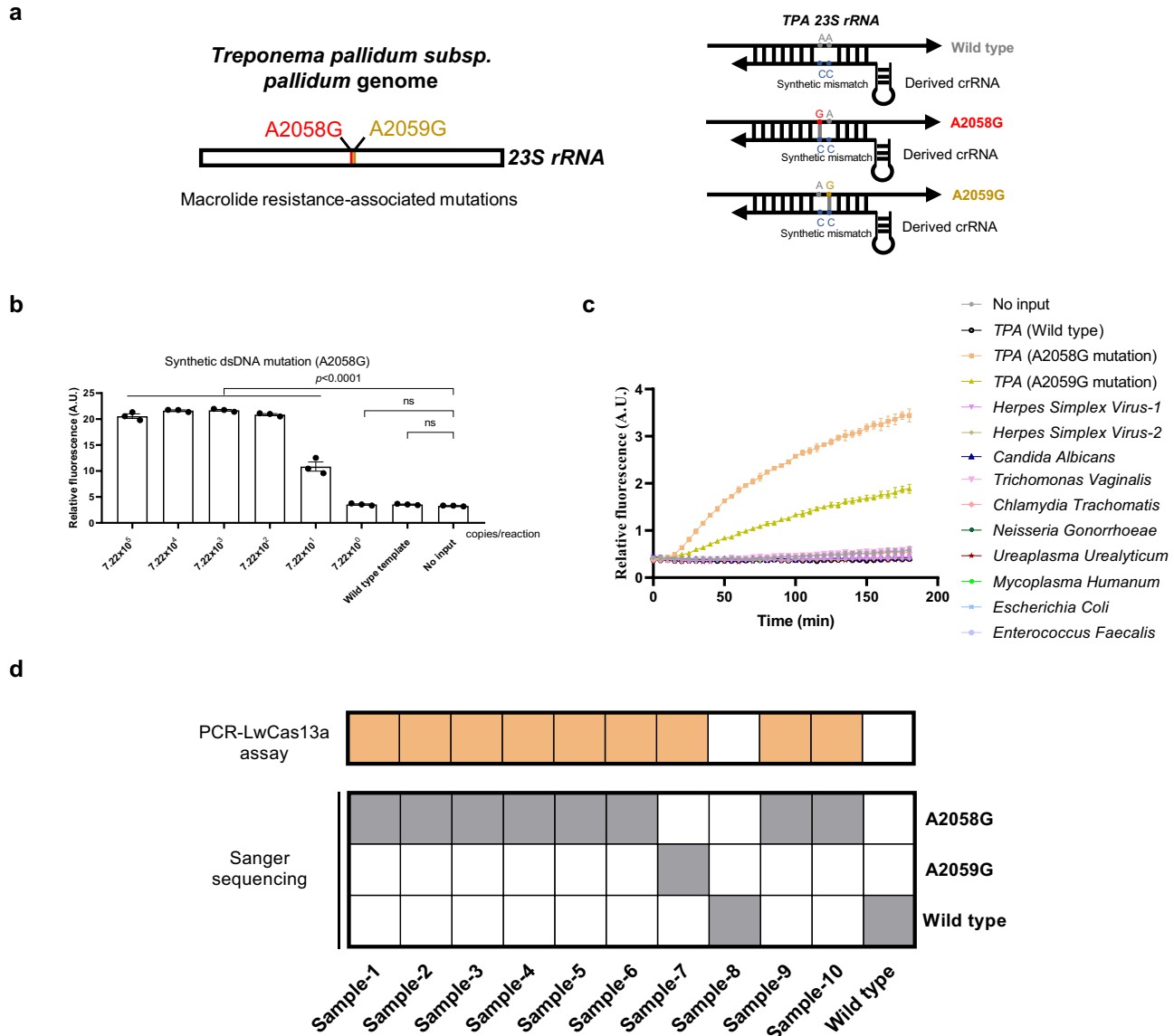

**Fig. 4 | PCR-LwCas13a assay for detection of azithromycin resistance.**
**a** Schematic of PCR-LwCas13a assay for *TPA* macrolide resistance genotyping.
**b** Dilution experiment to assess the limits of detection of *23 S rRNA* mutation
(A2058G). **c** Evaluation of the specificity of PCR-LwCas13a assay for *23 S rRNA*
mutations by testing a panel of DNAs from 10 genital microorganisms. **d** Ten clinical
samples were evaluated by PCR-LwCas13a assay and validated by Sanger

sequencing. Squares with color represent positive signals, while white squares
represent negative results. Abbreviations: A.U. Arbitrary units. *n* = 3 technical
replicates; two-tailed Student *t*-test was used to analyze the statistical significance;
*p* value was labeled in the figure, ns = not significant, error bars represent mean ±
SEM. Source data are provided as a Source Data file.

exceeded the performance of real-time PCR and nested PCR. The
major difference between the PCR-LwCas13a assay and other PCR-
based methods is cleavage of RNA reporters by the collateral activity of
LwCas13a, which enhances fluorescent signal and improves sensitivity.
Although a loop-mediated isothermal amplification (LAMP) assay
recently was developed for rapid diagnostic of *TPA*[55], the reported
limits of detection of LAMP (100 copies/reaction) are well below that
demonstrated herein for PCR-LwCas13a assay (single-copy/reaction).
Compared to other NAATs, PCR-LwCas13a assay is cheaper than
SHERLOCK and LAMP and only slightly more expensive than our in-
house TaqMan PCR (Supplementary Table 5), suggesting cost-
effectiveness for syphilis diagnosis in the future. RIT is considered
the gold standard for direct detection, with a sensitivity of
23 spirochetes[56] but is rarely performed because of technical com-
plexity and high cost. Our PCR-LwCa13a assay achieved greater sensi-
tivity than RIT in a clinical setting.

Nontreponemal test titers are widely used to evaluate the
response to syphilis treatment. However, about 20% of patients
become 'serofast' after therapy without clear evidence of treatment
failure[11–15], e.g., nontreponemal antibody titers that do not completely
revert to nonreactive after therapy despite an initial 4-fold decrease.
The PCR-LwCas13a assay was far more sensitive than real-time PCR[23]
and nested PCR[18] when applied to whole blood. To our knowledge, the
PCR-LwCas13a assay is the most sensitive direct assay for detection of
*TPA* DNA in whole blood reported to date. As such, the improved
sensitivity of our PCR-LwCas13a assay for whole blood could aid
interpretation of infection status when a 'serofast' state caused by low
*TPA* burden is suspected. The quantitative potential of PCR-LwCas13a
assay could be beneficial for treatment follow-up, however, further
investigation is still required to optimize this assay.

We further leveraged LwCas13a's high specificity to genotype *TPA*
strains without the need for costly and time-consuming sequencing

approaches. The ability to assign a clinical isolate to the Nichols- or SS14-like clade using PCR-LwCas13a could enhance surveillance and, more importantly, serves as proof-of-concept that evaluation of other phylogenetically informative loci with this assay is feasible. Because of multiple mismatches in the crRNAs for the *tp0548* sequences compared to the closely related pathogenic treponemes responsible for bejel and yaws, *Treponema pallidum subsp. endemicum* (*TEN*) and *Treponema pallidum subsp. pertenue* (*TPE*) (Fig. S6b), respectively, the genotyping assay is expected to have robust specificity for *TPA*. Recombination events have been documented to occur in the *tp0548* locus of *T. pallidum* subspecies[57,58]. These recombination events might negatively impact the ability of the PCR-LwCas13a assay to distinguish clades in rare cases. However, with two distinct crRNAs (one specific for the SS14 clade and the other specific for the Nichols clade), the genotyping assay is not expected to misidentify clades. In the case of a *TPA* strain with *TEN* recombination at the *tp0548* locus, we would be left with an indeterminate result (negative by SS14- and Nichols-specific PCR-LwCas13a assays) that could trigger additional investigation by sequencing. While the genotyping crRNAs were designed by screening the sequences from published *TPA*, *TPE*, and *TEN* strains, more rigorous validation with diverse clinical samples (including bejel) is needed to determine the role of PCR-LwCas13a for distinguishing *T. pallidum* subspecies. Our macrolide-resistance mutation assay provides an alternative to sequencing-based approaches that have limited utility in routine clinical practice due to slow turnaround times. Persons with significant allergic reaction to penicillin G require an alternative treatment for syphilis. While high prevalence of A2058G and A2059G *23S rRNA* mutations is established in many developed countries[27,28,36–38], recent reports of low prevalence in low- and middle-income countries (LMICs), such as Madagascar, Malawi, and Colombia, indicates preserved macrolide susceptibility in some locations[29]. Therefore, developing a rapid macrolide resistance genotyping method for syphilis, especially for pregnant women in LMICs, will be helpful. Our *tp0548* and *23S rRNA* genotyping assays provided results in 75–240 mins, with longer duration for samples expected to have low *TPA* burdens, that matched confirmatory Sanger sequencing.

Our approach has several limitations. First, the PCR-LwCas13a assay cannot be performed in a single reaction tube in its current form because the PCR and LwCas13a reactions require different temperatures. Second, our PCR-LwCas13a genotyping assays for lineage identification and drug-resistance assessment are less sensitive than our *TPA* detection assay. The latter successfully detected *TPA* *tpp47* target at an estimated single-molecule level. These losses in sensitivity are not unexpected because SNV-detection relies upon mismatch-designed crRNA selected from a restricted number of candidate crRNAs. Third, we did not evaluate the performance of our assays on some stages of syphilis (e.g., very early stage, latent, congenital syphilis or neurosyphilis). However, the large validation set of well-characterized and recently collected samples, including a subset subjected to RIT, provides a rare opportunity to evaluate assay performance across commonly collected clinical sample types. Fourth, the PCR-LwCas13a assay (75–240 min) takes more time than LAMP (15 min) and conventional qPCR (100 min). Despite the longer reaction time, our PCR-LwCas13a assay provides higher sensitivity and specificity than other NAATs for *TPA* diagnosis in clinical settings. Admittedly, the PCR-LwCas13a assay is more complicated than conventional PCR and, therefore, is not yet ready for routine clinical usage. However, towards this end, recently developed handheld detectors that combine microfluidics and PCR can achieve amplification and analysis of a target in 7.5–25 min[59,60]. These promising results raise the possibility that the PCR-LwCas13a assay will be utilized for point-of-care molecular diagnostics in the future. Due to the lack of a commercial PCR assay for detection of *TPA*, in-house qPCR was employed for comparison in the present study.

In conclusion, we developed a suite of PCR-LwCas13a syphilis assays with excellent sensitivity and specificity across multiple types of syphilis specimens. These assays have potential to overcome limitations of existing assays for *TPA* detection and offer a promising alternative to sequencing-based methods for molecular surveillance and drug-resistance genotyping.

## Methods

### Ethics statement and subjects
This study was approved by the Ethics Review Committee at Dermatology Hospital of Southern Medical University (GDDHLS-20181202[2 R], 2020056, 2021071). Patients were selected using a hospital-based convenience sampling approach. Eligible patients who visited the Dermatology Hospital of Southern Medical University were invited to participate from 2019 to 2021. The diagnosis of syphilis was made according to the guidelines of the STD Association, China Centers for Disease Control[61]. Primary syphilis cases with positive darkfield microscopy (DFM) tests and secondary syphilis cases with characteristic rashes and positive serological results (toluidine red unheated serum test [TRUST] and *Treponema pallidum* particle agglutination [TPPA]/treponemal chemiluminescence immunoassay [CIA]) were enrolled (Supplementary Table 4). Healthy volunteers were invited to join as negative controls; the negative serological status for syphilis of healthy volunteers was confirmed by CIA. Written informed consent was obtained from all participants. Following collection of research specimens, patients were treated according to the guidelines of the STD Association, China Centers for Disease Control (guidelines did not recommend azithromycin for any enrolled subjects)[61].

### Acquisition and processing of clinical samples
Sterile polyester swabs (Hcy technology, Shenzhen, China; CY-98000) were used to collect genital ulcer exudates from 36 primary syphilis cases (all positive by DFM) and 23 HSV-2 infected patients (clinical diagnosis and laboratory test confirmed), then placed into 250 μL of DNA/RNA Shield buffer (Zymo Research, Irvine, CA, USA; R1100–250) at −20 °C until DNA extraction per below. If RIT was conducted, two ulcer swabs were collected; the first was eluted into 1 mL TpCM-2 medium as previously described by Edmondson et al.[5] for RIT[7], and the second swab was placed in 250 μL of DNA/RNA Shield buffer and stored at −20 °C until DNA extraction. Thirty-five biopsies from secondary syphilis patients were collected by 4 mm punch and stored in 250 μL of DNA/RNA Shield buffer at −20 °C until DNA extraction. Eleven extracted DNAs of discarded surgical skin tissues from non-syphilis patients were set up as the non-syphilis-infected skin controls. Whole blood was collected by venipuncture from 64 syphilis patients and 47 volunteers, and 100 μL of whole blood was used for DNA extraction. RIT method for whole blood was described below. DNAs were extracted using the DNeasy Blood & Tissue Kit (QIAGEN, Hilden, Germany; 69506) according to the manufacturer's instructions; 100 μL of nuclease-free water was used to elute the DNA, followed by −80 °C storage until use. The DNAs from 10 genital microorganisms (Supplementary Table 3) were used to assess the specificity of the PCR-LwCas13a assay.

### Double-stranded DNA template preparation
The dsDNA templates (*tpp47* and *23S rRNA*) were amplified by using Q5 High-Fidelity DNA Polymerases (New England Biolabs, MA, USA; M0492S). A total of 25 μL reaction volume included 1.25 μL of 10 μM *tpp47*-dsDNA or *tp0548*-dsDNA or *23S rRNA*-dsDNA primers (Supplementary Table 6), 0.5 μL of 10 mM dNTP Mix, 0.25 μL of DNA polymerase, 1 μL of *TPA* gDNA extracted from rabbit-passaged Nichols or azithromycin-resistant clinical strains, and 20.75 μL of Invitrogen™ DNase/RNase-Free Distilled Water (Thermo Fisher Scientific, Waltham, MA, USA; 10977015). PCR was performed with the following conditions: 98 °C 30 s, and 40 cycles of 98 °C for 10 s, 62 °C for 20 s, and 72 °C for 30 s. 1.5% agarose gel was used to identify and purify the dsDNA template, and the target dsDNA was extracted by Universal

DNA purification kit (TIANGEN, Beijing, China; DP214-03) according to the manufacture's instruction. The purified dsDNA was quantified by Qubit dsDNA HS Assay Kit (Thermo Fisher Scientific, Waltham, MA, USA; Q33230). The Nichols and SS14 tp0548 dsDNAs were synthesized according to Nichols (GenBank ID: CP004010.2) and SS14 (GenBank ID: CP004011.1) reference genome. The series diluted aliquots of *tpp47*, *tp0548* and *23 S rRNA* dsDNA contained 1 ng/μL human DNA (extracted from HeLa cell line, CCDCC, Wuhan, China; GDC0009) to simulate human background in clinical samples.

### Determination of DNA concentration by digital droplet PCR
The concentrations of series diluted aliquots of dsDNA (*tpp47*) and 10 genital microorganisms DNA were evaluated by digital droplet PCR (ddPCR)[62]. Briefly, the ddPCR was performed using 20 μL of total reaction: 10 μL ddPCR Supermix for Probes (No dUTP) (BioRad, Hercules, CA, USA;186−3023), 0.9 μL of 10 μM *tpp47* primers (Supplementary Table 6), 0.5 μL of 10 μM probes, 1 μL of DNA template and 6.7 μL of Invitrogen™ DNase/RNase-Free Distilled Water (Thermo Fisher Scientific, Waltham, MA, USA; 10977015), followed by carrying on Droplet Digital PCR System (BioRad, Hercules, CA, USA; QX200) for generating droplets, sealing, PCR procedure with 95 °C 10 min, 45 cycles of 95 °C for 15 s and 60 °C for 1 min. The ddPCR primers used in this study are available in Supplementary Table 6.

### Expression and purification of LwCas13a
Expression and purification of LwCas13a were carried out according to the protocol developed by the Kellner et al.[63] with some modifications. LwCas13a plasmids (NovoPro Bioscience, Shanghai, China; V010159) were transformed into Rosetta (DE3) competent cells (Tiangen, Beijing, China; CB108) and incubated at 37 °C, 5% $CO_2$ for 16 h on LB Broth agar plate (Sangon Biotech, Shanghai, China; A507003) which contained 50 μg/ml Ampicillin (Sangon Biotech, Shanghai, China; A100339). Competent cells containing LwCas13a plasmids were grown at LB Broth media (Sangon Biotech, Shanghai, China; A507002) with 37 °C and 220 RPM until the value of $OD_{600}$ reached 0.6. Isopropylthio-β-galactoside (IPTG; Sangon Biotech, Shanghai, China; B541007) was added to media at 0.5 mM final concentration to induce protein expression for 4 h. Cells were then centrifuged at 12,000 x *g* for 1 min at 4 °C, and cell pellets were harvested and stored at −80 °C for further purification.

Protein purification was performed on ice. Cell pellets were resuspended and supplemented with protease inhibitors, lysozyme (1 mg/ml), and benzonase (5 μg/ml) followed by ultrasonic processor for crushing and purified with a His-tag Protein Purification Kit (Beyotime Biotechnology, Shanghai, China; P2226). Ultra-4 Centrifugal Filter Devices (Millipore, Darmstadt, Germany; UFC805096) were used to exchange the protein elution buffer with SUMO digestion buffer (30 mM Tris-HCl, 300 mM NaCl, 1 mM DTT, 0.2% NP-0.4, pH8.0) and LwCas13a protein was incubated at 25 °C for 3 h with SUMO protease (Novoprotein, Suzhou, China; PE007-01A). Following exchange of the SUMO digestion buffer with protein storage buffer (50 mM Tris, 600 mM NaCl, 5% Glycerol, 2 mM DTT, pH 7.5), the purified LwCas13a were stored at −80 °C for future use. All steps were analyzed and confirmed by SDS-PAGE and Coomassie Blue staining (Sangon Biotech, Shanghai, China; C510041). The concentration of protein was quantified by BCA Protein Assay Kit (Beyotime Biotechnology, Shanghai, China; P0012S).

### Preparation of crRNAs
To prepare crRNAs, oligonucleotides with an appended T7 promoter sequence, spacers (complementary to target RNA), and a crRNA core sequence (for binding to LwCas13a) were designed using SnapGene (v4.1.9) and synthesized by Sangon Biotech, Shanghai, China. Synthesized single-strand DNA (100 μM) was annealed to a short T7 primer (100 μM) previously described by Cunningham et al. and incubated at

37 °C overnight using the HiScribe T7 Quick High Yield RNA Synthesis kit (New England Biolabs, MA, USA; E2050S) to yield crRNA[45]. crRNAs were purified using RNAXP clean beads (Beckman, Brea, CA; A63987) with a 1.5x ratio of beads to reaction volume with manufacturer's instructions and quantified by Qubit RNA HS Assay Kit (Thermo Fisher Scientific, Waltham, MA, USA; Q32852). crRNAs were stored at −20 °C until use. All crRNAs used in this study are shown in Supplementary Table 6.

### PCR-LwCas13a assay
All PCR-LwCas13a primers used in this study are available in Supplementary Table 6. Primers (*tpp47*-T7, *tp0548*-T7, and *23 S rRNA*-T7) for PCR were designed using SnapGene (v4.1.9) and NCBI BLAST; for subsequent transcription, a T7 promoter was appended to the forward primer (Supplementary Table 6). Q5 High-Fidelity DNA Polymerase (New England Biolabs, MA, USA; M0491) was used to amplify all samples in the PCR step of PCR-LwCas13a assay. The reaction mixture (25 μL total volume) consisted of 1.25 μL of 10 μM primers, 0.5 μL of 10 mM dNTP Mix, 0.25 μL of DNA polymerase, variable volumes of input DNA, and variable volumes of nuclease-free water. PCR was performed using the following conditions: 98 °C 30 s, and 40 cycles for 98 °C 10 s, 60 °C 20 s, 72 °C 30 s. The input volume of DNA in the PCR-LwCas13a assay was 1 μL/reaction, except for whole blood samples (1.67 μL/reaction). 1.25 μL of PCR product was transferred to LwCas13a detection.

The LwCas13a reaction contains 40 mM Tris-HCl (pH 7.5), 9 mM $MgCl_2$, 1 mM rNTPs (New England Biolabs, MA, USA; N0466L), 2000 U/ml Murine RNase inhibitor (New England Biolabs, MA, USA; M0314L), 1500 U/ml T7 RNA Polymerase (New England Biolabs, MA, USA; M0251L), 225 nM crRNA, 45 nM purified LwCas13a and 125 nM collateral RNA reporter synthesized by Sangon biotech (5'/6-FAM-UUUUU-BHQ1/3') or RNaseAlert substrate (Integrated DNA Technologies, IDT, Coralville, IA, USA; 11-04-02-03). Synthesized RNA reporters were used in most experiments, except for the crRNA screen used in *TPA 23 S rRNA* mutation detection which used the IDT RNaseAlert substrate (Fig. S4). The reaction mixture was allowed to incubate for 3 h (except where indicated) at 37 °C on 96 Well Half-Area Microplate (Corning, NY, USA; CLS3694-100EA) with fluorescent kinetics measured at Ex/Em = 490 nm/520 nm every 5 min. RNA reporters are described in Supplementary Table 6. Three replicates were conducted for all PCR-LwCas13a assays.

### TaqMan PCR
TaqMan PCR was performed to compare with the PCR-LwCas13a assay in the detection of *tpp47* with different specimen sources. Primers and probes for *tpp47* are available in Supplementary Table 6 and synthesized by Sangon Biotech, Shanghai, China. The TaqMan PCR was performed using a total 25 μL reaction, including 12.5 μL of TaqMan Gene Expression Master (Thermo Fisher Scientific, Waltham, MA, USA; 4369016), 1.25 μL of 10 μM primers, 2 μL of 10 μM probe, 2 μL of 25 mM $MgCl_2$, DNA template and Invitrogen™ DNase/RNase-Free Distilled Water (Thermo Fisher Scientific, Waltham, MA, USA; 10977015), and measured on Real-Time PCR Instruments (BioRad, Hercules, CA, USA; CFX96) with following PCR procedure: 95 °C 10 min, 45 cycles for 95 °C 15 s and 60 °C 1 min. The input volume of DNA of TaqMan PCR in this study is 1 μL/reaction, except for the whole blood samples (1.67 μL/reaction). Three replicates were conducted for all TaqMan PCR.

### SHERLOCK
RPA primers and crRNA for *tpp47* are available in Supplementary Table 6. The SHERLOCK was performed as described by Cunningham et al.[45]. Briefly, TwistAmp Basic (TwistDx, Maidenhead, Berkshire, UK) was used to amplify all samples in the RPA step of the SHERLOCK assay. The reaction mixture (25 μL total volume) consisted of 1.2 μL of 10 μM

primers, 14.75 µL of rehydration buffer,1.25 µL of magnesium acetate, DNA template and Invitrogen™ DNase/RNase-Free Distilled Water (Thermo Fisher Scientific, Waltham, MA, USA; 10977015). The input volume of DNA in this study is 1 µL/reaction, except for the whole blood samples (1.67 µL/reaction). RPA was performed at 37 °C for 2 h. 1.25 µL of RPA product was transferred to LwCas13a detection. The LwCas13a detection was performed using a fluorescence microplate reader during 3 h of incubation as described above.

### Nested PCR for *tpp47*

Nested PCR primers for *tpp47* are available in Supplementary Table 6. Nested PCR was performed to compare with the PCR-LwCas13a assay in the detection of 30 whole blood samples. The reaction mixture (25 µL total volume) consisted of 0.5 µL of 10 µM primers, 0.5 µL of 10 mM dNTP Mix, 0.25 µL of DNA polymerase, 1.67 µL of DNA extracted from whole blood samples. The nested PCR amplification conditions were described by Grange et al.[18] and slightly modified as follows: 98 °C 3 min, 20 cycles (for first-round PCR) and 35 cycles (for second-round PCR) of 98 °C 1 min, 68 °C 30 s, and 72 °C 2 min followed by extension at 72 °C 10 min. The finial PCR products were analyzed by electrophoresis in 1.7% agarose gel as previously described[18]. Three replicates were conducted.

### Rabbit-infectivity testing

All animal experiments were approved by the Animal Welfare Committee of South China Agricultural University (2020c004) and conducted following the regulations of the institution. *TPA* strains were isolated by intratesticular inoculation of adult New Zealand White rabbits (male, 3 months of age, 2.5–3 kg) as previously described[7]. Briefly, each rabbit was tested for evidence of infection with *TPA* or *Treponema paraluiscuniculi* infection with TPPA before inclusion in this study. 1 mL of whole blood collected from a secondary syphilis patient or genital ulcer exudate diluted in 1 mL of TpCM-2 medium was injected into both testes of a rabbit housed at 18 °C. The rabbit's serologic status was monitored weekly by TPPA beginning with the second month after inoculation. Seropositive rabbits were sacrificed, and their testes were aseptically removed for *TPA* isolation. The RIT result was further confirmed by observing motile *TPA* in the testicular fluid under DFM. Genomic DNAs of isolated *TPA* strains were extracted using the DNeasy Blood & Tissue Kit (QIAGEN, Hilden, Germany; 69506). Seronegative rabbits were euthanized after 3 months of inoculation, followed by passaging the testicular fluid to a second rabbit. The continued passages for seronegative rabbits were conducted twice at most.

### Sanger sequencing and phylogenetic tree

*TPA 23 S rRNA* and *tp0548* genes were amplified by nested PCR using primers (*23 S rRNA*-external, *23 S rRNA*-internal, *tp0548*-external and *tp0548*-internal) (Supplementary Table 6) and sequenced by the Sanger as previously described[27]. A phylogenetic tree for *tp0548* sequences was generated using MEGA-X software (version 10.0.5) with the default parameters: Maximum Likelihood algorithm, Tamura-Nei model, and 1,000 bootstrapped replicates.

### Statistical analysis

Means and standard deviations were calculated by Prism 8 software (GraphPad, Inc., La Jolla, CA, USA). Mean differences in quantification were determined by Student's *t*-test. All statistical tests are two-sided, and samples with *p*-values < 0.05 are highlighted. Specimens were not all randomized, and experimenters were not blinded in conducting these experiments.

### Reporting summary

Further information on research design is available in the Nature Research Reporting Summary linked to this article.

## Data availability

The data supporting the findings of this study are available within the paper and supplementary information files. DNA sequencing data generated in this study have been deposited in the GenBank database under accession code ON776877.1-ON776886.1 and ON843723.1-ON843742.1. Nichols and SS14 reference genome were available in the GenBank database under accession code CP004010.2 and CP004011.1. Source data are provided with the paper.

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

## Acknowledgements

This study was funded by the Medical Scientific Research Foundation of Guangdong Province (A2018264 and B2019022 to W.C.), Guangdong Science and Technology Department (2017A020212008 to H.Z.), and the National Natural Science Foundation of China (81772240 to B.Y., 82072321 to W.K.). The study was also partially supported by the US National Institute of Allergy and Infectious Diseases (U19AI144177 to J.D.R. and R21AI148579 to J.B.P.). We thank Dr. Dongling Li from Southern Hospital of Southern Medical University for providing DNA stocks of genital microorganisms, Dr. Dingheng Zhu from Dermatology Hospital of Southern Medical University for assistance with the collection of non-syphilis skin control, Ms. Feifei Zhang, Ms. Daxiang Chen, Dr. Yaohua Xue, Dr. Rui Yue and Dr. Xiaomian Lin from Dermatology Hospital of Southern Medical University for assistance with patient recruitment and sample collection.

## Author contributions

H.Z. and B.Y. conceived the study. H.Z., B.Y., W.C., H.L., J.B.P., C.H.C., K.L.H. and J.D.R. designed the experiments. J.B.P. and C.H.C. provided experience with Cas13a-based diagnostic development and designed prototype experiments with W.C. and H.Z. W.C. and H.L. performed most of the experiments, data analysis, and data visualization. L.Z. and Y.P. performed DNA preparation, LwCas13a protein expression, and purification. Patient recruitment, sample collection, and the following care were carried out by W.K., L.Z., Y.P., W.C., J.Y., Y.J. and J.O., Y.J. performed RIT. W.C. and H.L. wrote the draft manuscript. W.C., H.L., L.Z., Y.P., J.B.P., Y.J., C.H.C., K.L.H., J.D.R., W.K., J.O., J.Y., B.Y., and H.Z. contributed to editing of the manuscript.

## Competing interests

J.B.P reports non-financial support from Abbott Diagnostics, grant support from Gilead Sciences and the World Health Organization, and an honorarium from Virology Education, all outside the scope of the current manuscript. J.D.R has licensing agreements with Biokit SA and ChemBio for recombinant *TPA* proteins for serodiagnosis of syphilis. All other authors declare no interest of conflicts.

## Additional information

**Peer review information** *Nature Communications* thanks other anonymous reviewer(s) to the peer review of this work. Peer review reports are available.

