## [Peer Review File · Nature Communications]

Novel suite of PCR-LwCas13a assays for detection and genotyping of *Treponema pallidum* in clinical samplesREVIEWER COMMENTS

Reviewer #1 (Remarks to the Author):

I have read with interest the manuscript: "Novel suite of PCR-LwCas13a assays for detection and genotyping of *Treponema pallidum* in clinical samples" by Wentao Chen and colleagues.

There has been a real need for rapid, reliable and sensitive assays able to detect treponemal DNA directly from clinical samples as well as rapid assays for macrolide resistance screening. Even though I think it is really good idea to use assays based on LwCas13a, I am convinced that the authors have to add additional experiments to prove the real importance of the LwCas13a in diagnosis and typing of pathogenic treponemes. In addition to this, I would like to raise a concern about the novelty of this study. Even though PCR-LwCas13 was never done before in the diagnostics and molecular typing of pathogenic treponemes, the same methodology was published before in numerous other pathogens. Moreover, most of these studies were developed as point-of-care testing that can be performed at a single temperature during a short time period.

Is there any chance that the authors can address the following issues?

- Cas13a (e.g. SHERLOCK) is usually combined with isothermal amplification. Isn't it worth to combine this assay with isothermal amplification in order to developed point-of-care pathogen detection? What would be the benefit of the potential LwCas13a assay when compared to the recently developed LAMP assay?

- One of the main discovery of the manuscript is that the PCR-LwCas13 assay achieved an order of magnitude better sensitivity then real-time PCR (i.e. 1 DNA copy per/ul) and, most importantly, the sensitivity of PCR-LwCas13 assay is significantly higher than qPCR in whole blood samples. Sensitivity of PCR based approaches heavily depends on optimization of the reaction. Storage and processing of the clinical samples is very important as well. Can the authors demonstrate this significant difference also with other sensitive PCR-based methods that are described and used? (for instance Nested-PCR, LAMP assay or qPCR with different primer design)

- Another important improvement of diagnosis and typing of pathogenic syphilis would be the introduction of inexpensive and rapid assay. Can the authors compared these two aspects of PCR-LwCas13 with other methods?

In addition:

- What is the sensitivity of the PCR-LWCas13a assay targeting the tp0548 locus in clinical samples? To my knowledge, the targeted short fragment of the tp0548 is not able to recognize the infection caused by other subspecies of *Treponema*, that can be found in misdiagnosed samples. Isn't it worth to choose other part of tp0548 or other locus that is able to distinguish Nichols vs SS14 as well as syphilis, yaws and bejel?

- What is the benefit of PCR-LwCas13 targeting the 23SrDNA in order to detect mutations responsible for macrolide resistance compared to the classical approach based on 23S rDNA amplification followed by RE digestion?

Reviewer #2 (Remarks to the Author):

The authors developed novel methods for molecular detection, genotyping, and identification of azithromycin resistance markers in *Treponema pallidum* using CRISPR-based assays. In light of the resurgence of syphilis worldwide, the authors should be commended for introducing a robust direct detection method for syphilis. It is clear from the authors' findings that their diagnostic method is superior to real-time PCR for detection of *T. pallidum*. Overall, the manuscript is well-written and easy to follow. I have minor comments/edits for the authors to consider.

Minor comments:

1.Line 63, Missing “-”? “38%~64%”

2.Line 160, “relied” should be changed to “relies”

3.Line 163, “EDCD” should be changed to “ECDC”

4.Line 274, Please indicate the type of swab (e.g, Dacron)

5.Line 277 & 283, “s” should be capitalized in “shield”

6.Line 388, Change “were” to “are”

7.Line 410, *Treponema pallidum* should be abbreviated to “TPA”

8.Figure 2, Define “RFU” in the legend.

9.The spelling of all bacterial species in the references should be corrected, e.g., the “t” in “*treponema pallidum*” should be capitalized and “*Chlamydia trachomatis*” is incorrectly spelt in reference #56.

Reviewer #3 (Remarks to the Author):

W. Chen and colleagues describe a novel PCR-LwCas13a assays for detection and genotyping of *Treponema pallidum* in clinical samples, a method which has recently been propagated also for other infectious diseases. The new test has been validated on 135 positive and 81 negative specimens and compared to normal qPCR and rabbit infectivity test. The new PCR had the highest sensitivity in this comparison. The test was then further adapted to screen for macrolide resistance and identify the TPA lineage.

The manuscript overall is very well written and qualifies for publication. It needs some minor revision.

Minor points

1. line 219ff: This whole paragraph is not clear in several ways
 - a. serofast needs proper definition, e.g. nontreponemal antibodies that do not completely revert to nonreactive after therapy despite initial 4-fold decrease.
 - b. Line 224ff: this is pure speculation because no data currently support this statement. This needs to be properly stated.
2. Line 276ff: this paragraph is inconsistent and not comprehensive:
 - a. For some samples numbers are given and for other not. Please give numbers for all samples.
 - b. When were the samples collected? Give the dates.
 - c. How were the patients selected, consecutive, by chance?
3. Can this new PCR method be done in a quantitative way? (Then it could be used for treatment follow-up, see also point X.b above)
4. Fig1A. For readers not familiar with the technique it is difficult to understand which steps are done and how many PCR reactions are performed in how many different tubes. Fig1A should be improved to give a better overview of the whole procedure, showing that pre-PCR and LwCas detection is done in different tubes. Further it should be shown in triplicate indicating the 3 targets for diagnosis, macrolide and typing.

5. Discussion: the downsides of the new procedure needs to be discussed, which is that this new test i) takes considerable longer than a normal qPCR, ii) is much more complicated incorporating different steps, iii) needs more knowhow and iv) as a consequence seems currently not be ready to be implemented in a normal diagnostic work-up

6. A further limitation should be stated; the assay has been compared to an in-house PCR and not to a commercial PCR.

Response letter

Dear Editor and Reviewers,

Thank you for your comments on our manuscript titled “Novel suite of PCR-LwCas13a assays for detection and genotyping of *Treponema pallidum* in clinical samples”. These comments helped us improve our manuscript. We have addressed the comments to the best of our abilities and revised the text to meet the requirements for publication.

We marked the revised portions in blue in the manuscript. The point-by-point responses to the reviewers’ comments are provided below:

Reviewer #1 (Remarks to the Author):

I have read with interest the manuscript: “Novel suite of PCR-LwCas13a assays for detection and genotyping of *Treponema pallidum* in clinical samples” by Wentao Chen and colleagues.

There has been a real need for rapid, reliable and sensitive assays able to detect treponemal DNA directly from clinical samples as well as rapid assays for macrolide resistance screening. Even though I think it is really good idea to use assays based on LwCas13a, I am convinced that the authors have to add additional experiments to prove the real importance of the LwCas13a in diagnosis and typing of pathogenic treponemes. In addition to this, I would like to raise a concern about the novelty of this study. Even though PCR-LwCas13 was never done before in the diagnostics and molecular typing of pathogenic treponemes, the same methodology was published before in numerous other pathogens. Moreover, most of these studies were developed as point-of-care testing that can be performed at a single temperature during a short time period.

Is there any chance that the authors can address the following issues?

- Cas13a (e.g. SHERLOCK) is usually combined with isothermal amplification. Isn't it worth to combine this assay with isothermal amplification in order to developed point-of-care pathogen detection? What would be the benefit of the potential LwCas13a assay when compared to the recently developed LAMP assay?

RESPONSE: We thank the reviewer for these comments. We agree that SHERLOCK (Cas13a combined with RPA) has the potential to be developed as point-of-care testing for pathogen

detection. However, in our preliminary testing using SHERLOCK, we found the performance of SHERLOCK for *TPA tpp47* detection to be inferior to that of PCR-LwCas13a assay. The PCR-LwCas13a assay (single-copy/reaction) achieved an order of magnitude better analytical sensitivity than SHERLOCK (10 copies/reaction) (see panel A, below) determined by comparison of both methods using serially diluted clinical samples (panel B). Moreover, we also compared the performance of PCR-LwCas13a assay, TaqMan PCR, and SHERLOCK using 10 whole blood samples from syphilis patients. Our PCR-LwCas13a assay detected *TPA* DNA in 9 of the 10 samples, while the TaqMan PCR and SHERLOCK assays were positive in only 5 and 6, respectively (panel C). Recently developed handheld detectors that combine microfluidics and PCR can achieve rapid (7.5-25 min) amplification and analysis of target (PMID: 25953325, 28624618). These promising results raise the possibility that the PCR-LwCas13a assay will be utilized for point-of-care molecular diagnostics in the future. A statement to this effect has been added on lines 280-283.

C

Sensitivity comparison among three *TPA* NAATs in whole blood samples

Whole blood	PCR-LwCas13a assay	TaqMan PCR	SHERLOCK
Sample-1	+	+	-
Sample-2	+	+	-
Sample-3	+	+	+
Sample-4	+	-	+
Sample-5	+	+	-
Sample-6	+	-	+
Sample-7	+	+	-
Sample-8	-	-	+
Sample-9	+	-	+
Sample-10	+	-	+
Sensitivity	90% (9/10)	50% (5/10)	60% (6/10)

LAMP was described as a rapid and sensitive isothermal amplification method that amplifies a limited amount of DNA copies into a million copies (PMID: 10871386). Recently a LAMP assay for

detection of *TPA* DNA was developed (PMID: 29649256); however, the reported limits of detection of LAMP (100 copies/reaction) are well below the sensitivity of the PCR-LwCas13a assay (single-copy/reaction) reported in our study. These findings are in accord with reported data showing that a recently developed RT-LAMP for SARS-CoV-2 diagnosis is less sensitive than conventional RT-PCR (PMID: 34385527, 34856308). Compared to LAMP assay, our PCR-LwCas13a assay is not only more sensitive, it also has robust specificity due to sequence-specific crRNA-target recognition. The major drawbacks of LAMP are the high risk of carryover contamination and the increased chance of false positive results due to primer–primer hybridizations caused by the use of multiple primers (PMID: 24577617, 24323513, 34376716). We point out as well that primer design for PCR-LwCas13a assay is straightforward while primer set design for LAMP is complicated and unavailable for some of targets. An additional advantage of our PCR-LwCas13a assay is that it can be employed for SNV-detection (e.g., drug-resistant genetic markers/lineage genotyping), while, to the best of our knowledge, LAMP for *TPA* genotyping has not been reported. These combined applications potentially provide valuable information for diagnosis, treatment, monitoring of therapeutic response, and epidemiologic surveillance.

On lines 111-118, we have added the following sentence: “We compared our assay to SHERLOCK, another sensitive CRISPR-based diagnostic that combines recombinase polymerase amplification (RPA) and CRISPR-LwCas13a detection^{40,42,43,45}. However, we found the performance of SHERLOCK for *TPA tpp47* detection to be inferior to that of PCR-LwCas13a assay. The PCR-LwCas13a assay (single-copy/reaction) achieved an order of magnitude better analytical sensitivity than SHERLOCK (10 copies/reaction) (Fig. 2a and S3a) determined by comparison of both methods using serially diluted clinical samples (Fig. S3b).”

On line 227 of the revised manuscript we have added a sentence relating to comparison of the PCR-LwCas13a assay and LAMP: “Although a loop-mediated isothermal amplification (LAMP) assay recently was developed for rapid diagnostic of *TPA*⁵⁵, the reported limits of detection of LAMP (100 copies/reaction) are well below that demonstrated herein for PCR-LwCas13a assay (single-copy/reaction).”

- One of the main discovery of the manuscript is that the PCR-LwCas13 assay achieved an order of magnitude better sensitivity than real-time PCR (i.e. 1 DNA copy per/ul) and, most importantly, the sensitivity of PCR-LwCas13 assay is significantly higher than qPCR in whole blood samples. Sensitivity of PCR based approaches heavily depends on optimization of the reaction. Storage and processing of the clinical samples is very important as well. Can the authors demonstrate this significant difference also with other sensitive PCR-based methods that are described and used?

(for instance Nested-PCR, LAMP assay or qPCR with different primer design)

RESPONSE: The major difference between PCR-LwCas13a assay and other PCR-based methods is the secondary amplification of the *TPA* target by T7 RNA polymerase in the former. Also, sequence-specific crRNA-target recognition ensures the specificity of the PCR-LwCas13a assay. Mismatches of crRNA to target would not activate LwCas13a to cleave RNA reporters. These modifications greatly improve the sensitivity and specificity of the new assay for *TPA* detection. To our knowledge, our new assay for *TPA tpp47* detection harbors the highest sensitivity when compared to nested-PCR (PMID: 29739928, 22219306) and LAMP assay (PMID: 29649256) as well as real-time qPCR using another widely used target (*poIA*) (PMID: 20502522, 32578865). The superior performance of PCR-LwCas13a assay highlights the potential of CRISPR-based approaches to improve syphilis diagnosis.

To clarify these points, we have added the following sentences to the revised manuscript:

Line 223: “The major differences between the PCR-LwCas13a assay and other PCR-based methods are the secondary amplification of the *TPA* target by T7 RNA polymerase, the sequence-specific crRNA-target recognition and the cleavage of an RNA reporter, which improve both sensitivity and specificity.”

Line 240: “The PCR-LwCas13a assay was far more sensitive than real-time PCR²³ and nested PCR¹⁸ when applied to whole blood.”

- Another important improvement of diagnosis and typing of pathogenic syphilis would be the introduction of inexpensive and rapid assay. Can the authors compare these two aspects of PCR-LwCas13 with other methods?

RESPONSE: Thank you for your suggestion.

We have provided the consumables cost calculation in **Supplementary Table 4**. The PCR-LwCas13a assay is cheaper than SHERLOCK and LAMP and just slightly more expensive than TaqMan PCR in this study.

On line 230 we have added the following sentence: “Compared to other NAATs, PCR-LwCas13a assay is cheaper than SHERLOCK and LAMP and just slightly more expensive than our in-house TaqMan PCR (**Supplementary Table 4**), suggesting its cost-effectiveness for syphilis diagnosis in the future.”

Regarding the rapidity of the assay:

In **Fig. 2a** and **Fig. S3** we have provided additional data to demonstrate the performance characteristics of the PCR-LwCas13a. The PCR-LwCas13a assay could detect all serial dilutions of *tpp47* dsDNA by a 75 min reaction (60 min pre-PCR pairing with 15 min LwCas13a).

On line 124 we have added the following sentence: "The PCR-LwCas13a assay can be optimized for rapid performance. As shown in **Fig. S4**, the assay could detect all serial dilutions of *tpp47* dsDNA within a 75 min reaction time (60 min pre-PCR pairing with 15 min LwCas13a)."

In addition:

- What is the sensitivity of the PCR-LwCas13a assay targeting the *tp0548* locus in clinical samples? To my knowledge, the targeted short fragment of the *tp0548* is not able to recognize the infection caused by other subspecies of *Treponema*, that can be found in misdiagnosed samples. Isn't it worth to choose other part of *tp0548* or other locus that is able to distinguish Nichols vs SS14 as well as syphilis, yaws and bejel?

RESPONSE: Thanks for your suggestion. We have performed additional experiments for evaluating the sensitivity of PCR-LwCas13a assay for *TPA* lineage identification in clinical samples. Within our testing panel of 33 *tpp47*-positive clinical samples, the PCR-LwCas13a assay successfully identified the *TPA* lineages of 32 specimens with a sensitivity of 96.9%.

On line 179 we have added the following sentence: " The PCR-LwCas13a assay successfully identified *TPA* lineages for 32 of 33 *tpp47*-positive clinical specimens, a sensitivity of 96.9% (**Fig. 3f**)."

We agree that other *Treponema* subspecies cannot be identified using the short fragment of *tp0548*. The genotyping assay depends upon binding of crRNA to the target gene and the crRNA-guided cleavage activity of LwCas13a. This means that only 'On' or 'OFF' target status can be employed for *TPA* lineage identification, as done for Nichols and SS14 genotyping herein. Although it is desirable to develop a CRISPR-based approach for distinguishing *Treponema* subspecies, it is beyond the scope of the present study. Given the resurgence of syphilis worldwide, we also believe CRISPR-based diagnosis and epidemiologic surveillance for *TPA* deserves priority. SNV-detection relies upon mismatch-designed crRNA selected from a restricted number of candidate crRNAs. Thus, crRNA designed under these challenging criteria would diminish assay sensitivity.

- What is the benefit of PCR-LwCas13 targeting the 23SrDNA in order to detect mutations responsible for macrolide resistance compared to the classical approach based on 23S rDNA amplification followed by RE digestion?

RESPONSE: The conventional approach for detecting genetic markers of macrolide resistance in *TPA* is time-consuming and low-throughput because it requires nested-PCR for 23S *rRNA* amplification followed by restriction enzyme digestion and agarose gel-based analysis. (PMID: 15247355, 27100768). Moreover, identification of the A2058G and A2059G mutations requires different restriction enzymes (MbolI and BsaI, respectively), and agarose gel-based analysis can be subjective. Our PCR-LwCas13a assay provides an objective, sensitive, high-throughput and rapid tool to identify macrolide-resistant genetic markers.

Reviewer #2 (Remarks to the Author):

The authors developed novel methods for molecular detection, genotyping, and identification of azithromycin resistance markers in *Treponema pallidum* using CRISPR-based assays. In light of the resurgence of syphilis worldwide, the authors should be commended for introducing a robust direct detection method for syphilis. It is clear from the authors' findings that their diagnostic method is superior to real-time PCR for detection of *T. pallidum*. Overall, the manuscript is well-written and easy to follow. I have minor comments/edits for the authors to consider.

Minor comments:

1.Line 63, Missing “-“? “38%~64%”

RESPONSE: Thank you for identifying the error. We have corrected it (line 58).

2.Line 160, “relied” should be changed to “relies”

RESPONSE: We have revised the sentence “... *TPA* genotyping relies on sequence-based approaches...” (line 168)

3.Line 163, “EDCD” should be changed to “ECDC”

RESPONSE: We have corrected the error. The sentence is now “...a locus widely used for ECDC and MLST genotyping of *TPA* strains...” (line 171)

4.Line 274, Please indicate the type of swab (e.g, Dacron)

RESPONSE: On line 310, we have indicated the type of swab in sentence “Sterile polyester swabs (Hcy technology, Shenzhen, China; CY-98000) were used ...”.

5.Line 277 & 283, “s” should be capitalized in “shield”

RESPONSE: “shield” has been corrected to “Shield” in lines 314 and 319.

6.Line 388, Change “were” to “are”

RESPONSE: Thanks. We have made the requested corrected this error (line 425).

7.Line 410, Treponema pallidum should be abbreviated to “TPA”

RESPONSE: We have corrected this error (line 457).

8.Figure 2, Define “RFU” in the legend.

RESPONSE: Thanks. We have discarded this word in the figure legend, because no “RFU” was used in the figure.

9.The spelling of all bacterial species in the references should be corrected, e.g., the “t” in “treponema pallidum” should be capitalized and “Chlamydia trachomatis” is incorrectly spelt in reference #56.

RESPONSE: We have corrected these errors in the references section.

Reviewer #3 (Remarks to the Author):

W. Chen and colleagues describe a novel PCR-LwCas13a assays for detection and genotyping of Treponema pallidum in clinical samples, a method which has recently been propagated also for other infectious diseases. The new test has been validated on 135 positive and 81 negative specimens and compared to normal qPCR and rabbit infectivity test. The new PCR had the highest sensitivity in this comparison. The test was than further adapted to screen for macrolide resistance

and identify the TPA lineage.

The manuscript overall is very well written and qualifies for publication. It needs some minor revision.

Minor points

1. line 219ff: This whole paragraph is not clear in several ways
 - a. serofast needs proper definition, e.g. nontreponemal antibodies that do not completely revert to nonreactive after therapy despite initial 4-fold decrease.
 - b. Line 224ff: this is pure speculation because no data currently support this statement. This needs to be properly stated.

RESPONSE: We thank the Reviewer for his/her comments. We have improved wording for this paragraph.

To define the serofast state, on line 239 we have added "...e.g., nontreponemal antibody titers that do not completely revert to nonreactive after therapy despite an initial 4-fold decrease."

We also improving the wording on line 243: "As such, the improved sensitivity of our PCR-LwCas13a assay for whole blood could aid interpretation of infection status when a 'serofast' state caused by low *TPA* burden is suspected."

2. Line 276ff: this paragraph is inconsistent and not comprehensive:
 - a. For some samples numbers are given and for other not. Please give numbers for all samples.
 - b. When were the samples collected? Give the dates.
 - c. How were the patients selected, consecutive, by chance?

RESPONSE: Thanks. We have provided the numbers for all samples (lines 318 and 322) and the dates of sample collection (line 297). On line 295 we now state that patients were selected using a hospital-based convenience sampling approach.

3. Can this new PCR method be done in a quantitative way? (Then it could be used for treatment follow-up, see also point X.b above)

RESPONSE: Thank you. The PCR-LwCas13a assay has the potential to be conducted quantitatively, unlike many LwCas13a-RPA (SHERLOCK) assays. A correlation ($R^2 = 0.897$) of the copy numbers of *TPA tpp47* synthetic DNA with detected fluorescence was observed following 15 min LwCas13a detection.

On line 126 we have added “We observed a correlation ($R^2 = 0.897$) of copy numbers of *TPA tpp47* synthetic dsDNA with detected fluorescence under 15 min LwCas13a detection (**Fig. S5**), suggesting the potential of the PCR-LwCas13a assay for DNA quantitation.”

On line 246 we have added “The quantitative potential of PCR-LwCas13a assay could be beneficial for treatment follow-up, however, further investigation is still required to optimize this assay.”

4. Fig1A. For readers not familiar with the technique it is difficult to understand which steps are done and how many PCR reactions are performed in how many different tubes. Fig1A should be improved to give a better overview of the whole procedure, showing that pre-PCR and LwCas detection is done in different tubes. Further it should be shown in triplicate indicating the 3 targets for diagnosis, macrolide and typing.

RESPONSE: Thank you. We have improved the schematic in Figure 1 to more clearly illustrate the procedural overview. The new schematic clearly shows that pre-PCR and LwCas13a detection are performed in separate tubes and that three targets of *TPA* were detected in separate reactions in triplicate for diagnosis, identification of lineage, and macrolide-resistant genetic markers, respectively.

5. Discussion: the downsides of the new procedure needs to be discussed, which is that this new test i) takes considerable longer than a normal qPCR, ii) is much more complicated incorporating different steps, iii) needs more knowhow and iv) as a consequence seems currently not be ready to be implemented in a normal diagnostic work-up.

RESPONSE:

Thanks for your suggestion.

On lines 275-283, we have addressed these limitations by adding the following: “Fourth, the PCR-LwCas13a assay (75-240 min) takes more time than LAMP (15 min) and conventional qPCR (100 min). Despite the longer reaction time, our PCR-LwCas13a assay provides higher sensitivity and specificity than other NAATs for *TPA* diagnosis in clinical settings. Admittedly, the

PCR-LwCas13a assay is more complicated than conventional PCR and, therefore, is not yet ready for routine clinical usage. However, towards this end, recently developed handheld detectors that combine microfluidics and PCR can achieve amplification and analysis of a target in 7.5-25 min^{57,58}. These promising results raise the possibility that the PCR-LwCas13a assay will be utilized for point-of-care molecular diagnostics in the future.”

6. A further limitation should be stated; the assay has been compared to an in-house PCR and not to a commercial PCR.

RESPONSE:

On line 283 in the revised manuscript, we have added “Due to the lack of a commercial PCR assay for detection of *TPA*, in-house qPCR was employed for comparison in the present study.”

REVIEWER COMMENTS

Reviewer #1 (Remarks to the Author):

Please find attached my comments to the responses for my comments in the first round:

Comment #1

Please accept my apology for not being able to make my comment clear. Point of care tests should be very simple, easy, fast and without the need of special equipment. Combining SHERLOCK-like diagnosis with isothermal amplification would enable the clinicians to read the results based on colorimetric reaction without the need of qPCR machine (and that is of course also connected with lower sensitivity compared to temperature – dependent amplification). But maybe this topic is out of the scope of the original manuscript.

Comment #2

The authors claims that PCR-LwCas13a has following advantages compared to PCR-based method:

- Higher specificity
- Higher sensitivity

I do appreciate that crRNA-target recognition ensure the specificity, however, this can be ensure with a good primer design when using conventional PCR methods. The results in the paper shows that PCR-LwCas13a is indeed great in reaching a very good sensitivity and is able to amplify treponemal DNA even from whole blood samples. However, as stated in my original comment, I do believe this has to be shown clearly by comparing the PCR-LwCas13a results with other PCR-based technique using the same sample set. The sensitivity can be also influenced by many other factors, for example, storage and processing of the clinical samples. Hence, comparing sensitivity of PCR-LwCas13a and sensitivity of other PCR-based methods previously published using different samples set, is not optimal. In addition, I do not agree with the author's statement that the major difference between PCR-LwCas13a and other PCR-based method based on the secondary amplification. Nested PCR is based on secondary amplification as well. PCR-LwCas13a could be superior to the nested PCR in this regard, only if PCR-LwCas13a targets ribosomal RNA (that is present in multiple copies in the bacterial cells), which is not the case of the diagnostic locus selected in this study.

Comment #4

Is there any specific reason why are you using rabbit-passed samples for TP0548 validation assay? (Line 176) Rabbit-passed samples usually contain several orders of magnitude more treponemal DNA than clinical samples. I think it would be fair to try out this assay on the real clinical samples. Also, using the term "clinical samples" when using rabbit-passed samples is not correct. Can you say how many copies

of treponemal DNA per ul is the TP0548 assay able to detect? In addition, given the increasing number of detected misdiagnosed cases of bejel (*T.p.endemicum*) with syphilis (*T.p.pallidum*), I do believe that choosing the fragment of the TP0548 able to distinguish different subspecies of pathogenic treponemes is crucial for correct epidemiological data.

REVIEWER COMMENTS

Reviewer #1 (Remarks to the Author):

We thank the Reviewer for their additional constructive comments regarding the first revision of our manuscript. Our responses are below:

Comment #1

Please accept my apology for not being able to make my comment clear. Point of care tests should be very simple, easy, fast and without the need of special equipment. Combining SHERLOCK-like diagnosis with isothermal amplification would enable the clinicians to read the results based on colorimetric reaction without the need of qPCR machine (and that is of course also connected with lower sensitivity compared to temperature – dependent amplification). But maybe this topic is out of the scope of the original manuscript.

RESPONSE:

Developing a point of care assay was our initial intention. However, in preliminary experiments, we found that the performance of SHERLOCK for detection of *tpp47* was inferior to the PCR-LwCas13a assay. These results have been added to the revised manuscript: lines 115-123, **Figures S3a, S3b**, and **Supplementary Table 1**. We agree that adapting PCR-LwCas13a assay for point of care usage is important, but it is beyond the scope of this manuscript.

Comment #2

The authors claims that PCR-LwCas13a has following advantages compared to PCR-based method:

- Higher specificity
- Higher sensitivity

I do appreciate that crRNA-target recognition ensure the specificity, however, this can be ensure with a good primer design when using conventional PCR methods. The results in the paper shows that PCR-LwCas13a is indeed great in reaching a very good sensitivity and is able to amplify treponemal DNA even from whole blood samples. However, as stated in my original comment, I do believe this has to be shown clearly by comparing the PCR-LwCas13a results with other PCR-based technique using the same sample set. The sensitivity can be also influenced by many other factors, for example, storage and processing of the clinical samples. Hence, comparing sensitivity of PCR-LwCas13a and sensitivity of other PCR-based methods previously published using different samples set, is not optimal. In addition, I do not agree with the author's statement that the major difference between PCR-LwCas13a and other PCR-based method based on the secondary amplification. Nested PCR

is based on secondary amplification as well. PCR-LwCas13a could be superior to the nested PCR in this regard, only if PCR-LwCas13a targets ribosomal RNA (that is present in multiple copies in the bacterial cells), which is not the case of the diagnostic locus selected in this study.

RESPONSE: We used the same samples to compare the sensitivity of the PCR-LwCas13a assay versus real-time PCR (**Fig. 2**) as well as versus ddPCR (**Fig. S2**). In lines 143-146 of the revision, we state that “For comparison, the *tpp47* PCR-LwCas13a assay and TaqMan PCR were conducted in parallel. The overall sensitivities for *TPA* detection were 93.33% (95% CI: 87.72%-96.91%) for the PCR-LwCas13a assay and 70.37% (95% CI: 61.91%-77.92%) for TaqMan PCR.” In addition, we compared the PCR-LwCas13a assay with nested PCR in 10 syphilis whole blood samples. In lines 123-125 of the revised manuscript, we state “For this convenience sample set, the PCR-LwCas13a assay also exhibited higher sensitivity than *tpp47*-based nested PCR (see **Supplementary Table 1**).”

For samples that contain low *TPA* burden samples (e.g., whole blood), the sensitivity of TaqMan PCR (43.75%) and nested PCR (60%) in our study were not significantly different from values (13%-64%) reported previously (Grange *et al*, J Clin Microbiol, 2012; Cruz *et al*, PLoS Negl Trop Dis, 2010). These comparable results established a “baseline” for evaluation of the performance of the PCR-LwCas13a assay, which outperformed both conventional approaches.

Regarding our statement about secondary amplification, we intended to highlight the unique mechanism of signal amplification offered by the PCR-LwCas13a assay. Unlike PCR, LwCas13a exhibits collateral activity after recognition and cleavage of a target transcript, leading to non-specific degradation of any nearby transcripts as well as RNA reporters. Consequently, one activated LwCas13a protein will cleave more than one molecule of RNA reporter, leading to amplification of fluorescence signal. This effect is likely responsible for the increased sensitivity of the PCR-LwCas13a assay. We have corrected the wording of lines 233-236 in the revised manuscript as follows: “The major difference between the PCR-LwCas13a assay and other PCR-based methods is cleavage of RNA reporters by the collateral activity of LwCas13a, which enhances fluorescent signal and improves sensitivity.”

Comment #4

Is there any specific reason why are you using rabbit-passed samples for TP0548 validation assay? (Line 176) Rabbit-passed samples usually contain several orders of magnitude more treponemal DNA than clinical samples. I think it would be fair to try out this assay on the real clinical samples. Also, using the

term “clinical samples” when using rabbit-passed samples is not correct. Can you say how many copies of treponemal DNA per ul is the TP0548 assay able to detect? In addition, given the increasing number of detected misdiagnosed cases of bejel (*T.p.endemicum*) with syphilis (*T.p.pallidum*), I do believe that choosing the fragment of the TP0548 able to distinguish different subspecies of pathogenic treponemes is crucial for correct epidemiological data.

RESPONSE: We thank the Reviewer for this comment. We agree on the need to include clinical samples in our validation of the *tp0548* genotyping assay. Thus, the revised manuscript now shows results for ten skin biopsy samples and their corresponding rabbit passed isolates. As shown below, for all samples, the lineages of *TPA* strains determined by PCR-Cas13a assay matched the confirmatory Sanger sequencing.

On lines 183-189 of the revised manuscript, we have added the following sentences: “To validate this genotyping assay, we compared the traditional molecular typing approach (PCR followed by Sanger sequencing) with the PCR-LwCas13a assay using DNA extracted from ten skin biopsy samples and their corresponding rabbit-passaged isolates (twenty total samples). Results of the PCR-LwCas13a genotyping assay matched the Sanger sequencing clade assignments for all SS14- and Nichols-like samples tested (**Fig. 3f**), thereby confirming that the PCR-LwCas13a assay is capable of *TPA* genotyping.”

We validated the analytical sensitivity of the genotyping assay using synthesized *tp0548* dsDNA (Nichols and SS14 strains) in dilution series in which the mock clinical samples contained 1 ng human DNA (see figure, below). On lines 181-183 of the revised manuscript we have added the following sentence: “In dilution series using synthesized *tp0548* DNA for the Nichols and

SS14 strains, the assay yielded genotyping data for samples containing as few as 10 copies per reaction (Fig. 3e).”

We apologize for not clearly pointing out that the Nichols- and SS14-lineage crRNAs used for TPA genotyping do not target *tp0548* sequences in the closely related pathogenic treponemes *T. pallidum* subsp. *pertenue* and *T. pallidum* subsp. *endemicum* (see figure below), although the PCR primers do amplify the corresponding *tp0548* fragments. Consequently, the PCR-LwCas13a assay is expected to have robust specificity for TPA. In lines 261-266 of the revision, we have added the following sentence: “Because of multiple mismatches in the crRNAs for the *tp0548* sequences compared to the closely related pathogenic treponemes responsible for bejel and yaws, *Treponema pallidum* subsp. *endemicum* and *Treponema pallidum* subsp. *pertenue* (Fig. S6b), respectively, the genotyping assay is expected to have robust specificity for TPA; additional testing on bejel and yaws clinical samples will be needed to determine the assay’s clinical specificity.”

REVIEWERS' COMMENTS

Reviewer #1 (Remarks to the Author):

I do believe that PCR-LWCas13a is going to be a great improvement of current syphilis diagnostics. However, the researchers have to be fully convinced that this is the case. I am happy that the authors followed my recommendation and compared the sensitivity of PCR-LWCas13a and nested PCR which is widely used for diagnostics of clinical samples that contain limited amount of treponemal DNA. However, I was disappointed that they selected only 10 samples to perform this experiment. How was this sample set selected? Is there any chance that the authors can prove the sensitivity of PCR-LWCas13a using the complete set of samples?

I am also not convinced that the selected fragment of the TP0548 for distinguishing the SS14 and Nichols strains is a wise option. Using this fragment, It is not possible to distinguish bejel and yaws samples. Bejel have been documented several time to be strongly mimicking clinical manifestation of syphilis. Moreover, TP0548 have been documented to be recombinant between bejel and syphilis strains, e.g. Nichols strains can carry TP0548 variant belonging to bejel and vice versa.

Otherwise I think that the manuscript is very good and the idea to use the PCR-LWCas13a is amazing.

REVIEWERS' COMMENTS

Reviewer #1 (Remarks to the Author):

We thank the Reviewer for additional comments.

I do believe that PCR-LWCas13a is going to be a great improvement of current syphilis diagnostics. However, the researchers have to be fully convinced that this is the case. I am happy that the authors followed my recommendation and compared the sensitivity of PCR-LWCas13a and nested PCR which is widely used for diagnostics of clinical samples that contain limited amount of treponemal DNA. However, I was disappointed that they selected only 10 samples to perform this experiment. How was this sample set selected? Is there any chance that the authors can prove the sensitivity of PCR-LWCas13a using the complete set of samples?

Response: To address the reviewer's concern, we compared the sensitivity of the PCR-LwCas13a assay and nested PCR for 20 additional whole blood samples (making a total of 30). The PCR-LwCas13a assay achieved a sensitivity of 83% compared to only 63% for nested PCR (see Table below and Supplementary Table 2). The sensitivity of nested PCR in our study is almost identical to the value (64%) reported previously (Cruz *et al*, PLoS Negl Trop Dis, 2010). These results for this expanded dataset are noted in lines 123-125 of the revised manuscript. Thus, in the present study, we have demonstrated the robustness of the PCR-LwCas13a assay by comparison with real-time PCR, nested PCR, SHERLOCK, and ddPCR.

Sensitivity comparison between PCR-LwCas13a assay and nested PCR in 30 whole blood samples.		
Sample ID	PCR-CRISPR	Nested PCR
Sample-1	+	+
Sample-2	+	+
Sample-3	+	-
Sample-4	+	-
Sample-5	+	-
Sample-6	+	-
Sample-7	-	+
Sample-8	+	+
Sample-9	+	+
Sample-10	+	+
Sample-11	-	+
Sample-12	+	+
Sample-13	+	+
Sample-14	+	+
Sample-15	+	-
Sample-16	+	-
Sample-17	+	+
Sample-18	+	+
Sample-19	-	-
Sample-20	+	+
Sample-21	+	+
Sample-22	-	-
Sample-23	+	-
Sample-24	+	+
Sample-25	+	-
Sample-26	-	+
Sample-27	+	+
Sample-28	+	+
Sample-29	+	+
Sample-30	+	-
Sensitivity	83.33%	63.33%

I am also not convinced that the selected fragment of the TP0548 for distinguishing the SS14 and Nichols strains is a wise option. Using this fragment, It is not possible to distinguish bejel and yaws samples. Bejel have been documented several time to be strongly mimicking clinical manifestation of syphilis. Moreover, TP0548 have been documented to be recombinant between bejel and syphilis strains, e.g. Nichols strains can carry TP0548 variant belonging to bejel and vice versa.

Response:

We agree with the Reviewer's comment that bejel strains sometimes can elicit clinical manifestations very similar to those of venereal syphilis. However, in a routine STD clinical context, the incidence of bejel is exceedingly rare so distinguishing infection by *Treponema pallidum subsp. pallidum* and *T. pallidum subsp. endemicum* is not a major diagnostic issue.

The reviewer is right to point out that recombination events have been documented to occur in the *tp0548* locus (Pla-Díaz M, *et al.*2022; Mikalová L, *et al.*2017; etc). It is possible that these recombination events might negatively impact our ability to distinguish clades in rare cases. However, because we employ two distinct crRNAs in separate assays – one specific for SS14 clade and one specific for Nichols clade – we do not expect to misassign the clade.

Rather, in the case of a *TPA* strain with *TEN* recombination at the *tp0548* locus, we would be left with an indeterminate result (negative by SS14- and Nichols-specific PCR-LwCas13a assays) that could trigger additional investigation by sequencing.

The genotyping assay in this study was intended to serve as “proof-of-principal” that distinguishing *TPA* clades is possible. It targets *tp0548* because this locus is widely used for ECDC and MLST genotyping of *TPA* strains. Though the genotyping crRNAs were designed by screening the sequences from published *TPA*, *TPE*, and *TEN* strains, more rigorous validation with diverse clinical samples (including bejel) is needed to determine the role of PCR-LwCas13a for distinguishing subspecies. This work is beyond the scope of the current manuscript but is a promising avenue of research that we plan to undertake.

On lines 265-276 of the revised manuscript, we have added the following sentences: “Recombination events have been documented to occur in the *tp0548* locus of *T. pallidum* subspecies^{57,58}. These recombination events might negatively impact the ability of the PCR-LwCas13a assay to distinguish clades in rare cases. However, with two distinct crRNAs (one specific for the SS14 clade and the other specific for the Nichols clade), the genotyping assay is not expected to misidentify clades. In the case of a *TPA* strain with *TEN* recombination at the *tp0548* locus, we would be left with an indeterminate result (negative by SS14- and Nichols-specific PCR-LwCas13a assays) that could trigger additional investigation by sequencing. While the genotyping crRNAs were designed by screening the sequences from published *TPA*, *TPE*, and *TEN* strains, more rigorous validation with diverse clinical samples (including bejel) is needed to determine the role of PCR-LwCas13a for distinguishing *T. pallidum* subspecies.”.

Otherwise I think that the manuscript is very good and the idea to use the PCR-LwCas13a is amazing.

We thank Reviewer for these constructive comments and are deeply gratified that he/she appreciates our work.